# Unveiling the Accuracy of Ultrasonographic Assessment of Thyroid Volume: A Comparative Analysis of Ultrasonographic Measurements and Specimen Volumes

**DOI:** 10.3390/jcm12206619

**Published:** 2023-10-19

**Authors:** Can Konca, Atilla Halil Elhan

**Affiliations:** 1Department of General Surgery, Ankara University School of Medicine, 06230 Ankara, Turkey; 2Department of Biostatistics, Ankara University School of Medicine, 06230 Ankara, Turkey; ahelhan@yahoo.com

**Keywords:** thyroid, ultrasonography, thyroid volume, thyroidectomy, ellipsoid formula, specimen volume

## Abstract

In endocrine surgery, a precise ultrasonographic measurement of thyroid volume is crucial. However, there is limited comparative research between ultrasonographic and specimen volumes, which has left this issue open to debate. This study aims to assess the accuracy of recommended formulas for ultrasonographic thyroid volume measurement by comparing them to specimen volumes and analyzing the influencing variables. From the data of 120 eligible patients, different formulas, including ultrasonographic thyroid volume (US-TV) based on the ellipsoid formula, lower correction factor thyroid volume (LCF-TV), and calculated ultrasonographic (derived formula) thyroid volume (CU-TV), were used to estimate the thyroid volume based on measurements taken prior to surgery. These measurements were compared with the intraoperative specimen volume (IO-TV) derived using Archimedes’ principle. According to our findings, the mean values for US-TV and LCF-TV were significantly lower, whereas CU-TV was higher than IO-TV. Deviations were more significant in patients who had surgery for benign indications or compressive symptoms and in those with suppressed thyroid-stimulating hormone levels. Although the ellipsoid formula tends to underestimate the actual thyroid volume, it remains the most accurate method for measuring ultrasonographic thyroid volume. The deviation is greater for larger volumes.

## 1. Introduction

Thyroid ultrasound is the gold standard imaging modality for evaluating the thyroid gland in contemporary medicine [1]. It is favored over other techniques such as computed tomography (CT) and magnetic resonance imaging (MRI) because of its non-invasive nature, cost-effectiveness, and availability [1,2,3]. Beyond its diagnostic use, ultrasound also measures the thyroid gland’s volume in two dimensions (2D), aiding in devising medical treatment strategies and choosing surgical approaches for patients [2,4]. For example, in minimally invasive thyroid surgery, a volume of 25 mL is seen as a cutoff for patient selection [5]. Moreover, removing intact thyroid specimens larger than 30 mL is expected to be challenging in the transoral endoscopic thyroidectomy vestibular approach (TOETVA), which is becoming increasingly popular among endoscopic thyroidectomy techniques [6]. On the other hand, it is traditionally believed that a larger thyroid volume is associated with increased rates of postoperative complications [7].

The volumetric evaluation of the thyroid gland with ultrasound is performed using the ellipsoid formula [3]. According to this formula, the mediolateral diameter (ML) × anteroposterior diameter (AP) × craniocaudal diameter (CC) is multiplied by π/6, yielding the value 0.524. However, a 1981 cadaveric study by Brunn et al. suggested that using a correction factor of 0.479 rather than 0.524 provided more accurate results [8].

In studies that evaluated the consistency of 2D thyroid volume measurement using ultrasound, various imaging methods, such as CT, MRI, and scintigraphy, have been used for comparison [3,4,9]. Although some studies have demonstrated that CT and MRI imaging provide more accurate results for thyroid volume measurements than ultrasound, the convenience and diagnostic value of ultrasound in assessing the thyroid gland make its practical applicability in daily practice unquestionable [1]. However, only a few studies have compared these radiologic calculations with surgical specimen measurements, which more accurately represent the actual thyroid volume. Three studies using Archimedes’ principle for specimen measurements have shown that ultrasonographic volume measurement underestimates the actual thyroid volume, with this discrepancy being attributed to the presence of multinodularity and a nodular isthmus [10,11,12]. Based on the evaluation of 92 patients, Trimboli et al. proposed an alternative volume calculation method based on a modification of the ellipsoid formula [10]. Yet, very few studies have affirmed the consistency of these formulas, indicating a need for further investigation in this domain.

The purpose of this study was to evaluate the accuracy of thyroid volume measurements, obtained using 2D ultrasound and various formulas, by comparing them to the actual volumes measured using the Archimedes’ principle on surgical specimens in light of recent technological advances in ultrasonography. We also aimed to identify factors influencing the consistency of these measurements and to devise a mathematical formula based on this assessment to yield more accurate results.

## 2. Materials and Methods

This study retrospectively analyzed 204 patients who underwent total thyroidectomy performed by a single surgeon (CK) at a tertiary care center between January 2020 and May 2022. The study received approval from the Ethical Review Committee of Ankara University School of Medicine (approval number: I03-162-23, approval date: 5 April 2023). We retrospectively included all patients who underwent total thyroidectomy by a single surgeon for the presence of preoperative cytological suspicion for malignancy or benign conditions such as Graves’ disease or multinodular goiter with compression symptoms, and who had data available for an ultrasonographic evaluation of the thyroid gland at the beginning of surgery, along with measurement of the volume of the thyroid specimen. Subsequently, patients were excluded from the study if they had retrosternal goiter or ectopic thyroid tissue that could affect the accuracy of the ultrasonographic size evaluation and volume measurements due to limited scanning depth or the presence of multifocal thyroid tissue. Additionally, those who underwent intraoperative cyst aspiration, had cyst rupture or compromised specimen integrity during dissection, exhibited isthmus nodularity, or lacked data for the relevant parameters were also excluded. These exclusions were necessary, as those factors could result in discrepancies between ultrasonographic and surgical volume calculations that would not be merely explained by the variation in measurement technique. In total, 120 patients who met the study criteria were analyzed. Patient details such as age, gender, body mass index (BMI), thyroid-stimulating hormone (TSH), thyroglobulin levels, anti-thyroglobulin antibody (anti-Tg Ab) levels, anti-thyroid peroxidase antibody (anti-TPO) levels, the indication for surgery, and pathological diagnoses were documented and assessed for this study.

This section outlines the details of the data collected through the clinical routines of the operating surgeon and the specifics of the surgical procedure and volume measurements. The preoperative diagnostic ultrasonography was performed in accordance with the EU-TIRADS classification system as part of the diagnostic evaluations of the patients [13]. Accordingly, EU-TIRADS 2 nodules with compressive symptoms, EU-TIRADS 3 nodules larger than 20 mm, EU-TIRADS 4 nodules larger than 15 mm, and all EU-TIRADS 5 nodules were evaluated through a fine-needle aspiration biopsy. Surgical indications were determined based on the cytological evaluation of nodules, according to the Bethesda classification system [14]. Patients with two consecutive Bethesda 3 category results or those categorized as Bethesda 4, 5, and 6 were considered to have a “suspicion for malignancy”. In Graves’ disease patients who did not meet the aforementioned biopsy criteria, surgical treatment was performed without a fine-needle aspiration biopsy. After obtaining informed consent from all patients post-diagnostic evaluation and preoperative preparation, an experienced endocrine surgeon skilled in thyroid ultrasonography conducted thyroid ultrasounds on patients scheduled for total thyroidectomy. These patients were positioned for the operation in the operating room under anesthesia (Figure 1). Measurements of the craniocaudal, mediolateral, and anteroposterior dimensions of each thyroid lobe were taken and documented at their maximal points (Figure 2). An 8–12 MHz linear transducer (Versana Active^®^, GE Healthcare Inc., Chicago, IL, USA) with a 38.4 mm imaging area was used for ultrasonographic measurements.

All patients underwent an open total thyroidectomy using a capsular dissection technique with a cervicotomy (Figure 3). After the dissection, the volume was determined by submerging the specimen in a graduated cylinder filled with an appropriate amount of saline enough to sink the specimen; the rise in the water level was noted and the difference was measured in milliliters, in line with Archimedes’ principle (Figure 4). The measurements of the intraoperative thyroid volume (IO-TV) were documented in milliliters (mL). As part of the standard clinical follow-up for total thyroidectomy patients, postoperative ultrasonography was performed one month after the surgery to ensure no residual tissue remained.

Following data collection, thyroid volume calculations were based on the measurements from all the patients, detailed below.

The thyroid volume derived from the ultrasound dimensions was termed as “Ultrasonographic Thyroid Volume (US-TV)”. The ellipsoid formula was utilized for US-TV calculation. Below is the ellipsoid formula used:$$Volume\left(\mathrm{mL}\right)=ML\left(\mathrm{cm}\right)\times AP\left(\mathrm{cm}\right)\times CC\left(\mathrm{cm}\right)\times 0.524$$

The results from the formula, calculated for each thyroid lobe separately, were aggregated to compute the total thyroid volume.

Calculations using the correction factor of 0.479, as proposed by Brunn et al., in place of the 0.524 correction factor in the ellipsoid formula, were labeled “Lower Correction Factor Thyroid Volume (LCF-TV)” [8]. Additionally, calculations using Trimboli et al.’s formula for thyroid volume measurement, expressed as “Volume = 1.24 × 2D-US-TV + 3.66”, were noted as “Calculated Ultrasonographic Thyroid Volume (CU-TV)” [10].

### Statistical Analysis

Descriptive statistics were summarized as counts and percentages for categorical variables, means, and standard deviations for normally distributed continuous variables, and medians with 25th and 75th percentiles for ordinal or non-normally distributed continuous variables. The Mann–Whitney U test was used for non-normally distributed continuous variables when there were two groups. Differences among three groups for non-normally distributed continuous variables were evaluated by Kruskal–Wallis variance analysis. A paired t-test was used to evaluate the differences between the methods compared. The limit of agreement was calculated using the Bland–Altman method. The degree of association between continuous variables was assessed using Pearson’s correlation coefficient and the concordance correlation coefficient (CCC). McBride’s strength of agreement criteria for this CCC (ρc) were used to assess the degree of equivalence between the two methods as follows: <0.90, poor; 0.90–0.95, moderate; 0.95–0.99, substantial; >0.99, almost perfect. The dependence of one outcome variable on two or more variables was evaluated using multiple linear regression analysis. Significance was set at *p* < 0.05. The R programming language (version 4.2.0) was used for the statistical analysis.

## 3. Results

The study comprised 120 patients (99 females and 21 males) with a mean age of 46.02 ± 13.2 years (range 17–75). The indication for surgery was suspicion for malignancy in the majority of the cases (83.3%), whereas the final pathologic assessment revealed a malignant diagnosis in 83 (69.2%) patients. When the details of the patients’ specimen pathologies were examined, the distribution of cancer subtypes in those with malignancies was found to be 78 (65%) cases of papillary, 2 (1.7%) cases of follicular, and 3 (2.5%) cases of medullary cancers, respectively. Among the 37 (30.8%) patients whose final pathology report indicated benign conditions, 31 (25.8%) of them were diagnosed with nodular goiter, whereas 6 (5%) patients were diagnosed with Graves’ disease. Among the 12 (10%) patients with Graves’ disease in the study, papillary cancer coexisted with Graves’ disease in the remaining 6 (5%) cases. The descriptive characteristics, the calculated ultrasonographic thyroid volumes, and the intraoperative thyroid volumes of the patients are presented in Table 1. At the postoperative follow-up, except for one patient with papillary carcinoma who had preoperative vocal cord paralysis due to recurrent laryngeal nerve invasion, none of the patients experienced transient or permanent recurrent laryngeal nerve paralysis when postoperative complications were investigated. Protracted hypoparathyroidism that resolved within the first postoperative month was seen in 21 patients, whereas 6 patients developed transient hypoparathyroidism that resolved within the first year after surgery. None of the patients experienced permanent hypoparathyroidism or postoperative neck hematoma.

### 3.1. Comparison of Calculated Volumes with Specimen Volume (IO-TV)

The mean volume measurements for IO-TV, US-TV, LCF-TV, and CU-TV were found to be 28.90 ± 24.96 mL, 26.15 ± 21.58 mL, 23.92 ± 19.74 mL, and 36.09 ± 26.76 mL, respectively. When the calculated thyroid volumes were compared to IO-TV, both US-TV (*p* = 0.005) and LCF-TV (*p* < 0.001) were significantly lower, whereas CU-TV (*p* < 0.001) was significantly higher than IO-TV (Table 2).

Compared with the thyroid specimen volumes measured by Archimedes’ principle (IO-TV), US-TV underestimated the volume in 60.8% (73/120) of the thyroid glands and overestimated the volume in 39.2% (47/120). LCF-TV underestimated the volume in 70.8% (85/120) and overestimated in 29.2% (35/120). CU-TV underestimated the volume in 18.3% (22/120) and overestimated in 81.7% (98/120) of the thyroid glands (Figure 5).

The Bland–Altman plots of the calculated volume measurements versus the specimen volumes are depicted in Figure 6. The limits of agreement (Mean ± 1.96 SD) were narrowest for US-TV (−23.6 and 18.1 mL). Of the 120 patients, 9 (7.5%) had US-TV values outside the limits of agreement, with 5 showing underestimation and 4 overestimation.

The correlation coefficient (r) between IO-TV and US-TV volumes was 0.905 (Table 3). Based on McBride’s strength of agreement criteria for CCC, the degree of equivalence between all comparisons with IO-TV was categorized as poor (Table 3).

### 3.2. Mathematical Formula Derived from US-TV Calculation for the Estimation of IO-TV

All mathematical formulas used to calculate thyroid volume from ultrasonographic measurements demonstrated a strong correlation with IO-TV (Table 3). According to the linear regression analysis, US-TV, LCF-TV, and CU-TV had the highest R^2^ values with IO-TV (R^2^ = 0.820 for all). The mathematical formula obtained from the linear regression analysis between IO-TV and US-TV was as follows (Figure 7):$$\left(\mathrm{IO}-\mathrm{TV}\right)=1.522+1.047\times \left(\mathrm{US}-\mathrm{TV}\right)$$

### 3.3. Analysis of the Volume Differences between Calculated Thyroid Volumes and Specimen Volumes

The differences between the calculated thyroid volumes and the IO-TV are detailed in Table 2. The deviation from underestimation in calculations compared to IO-TV was −2.75 ± 10.65 mL for US-TV, whereas it was −4.98 ± 10.98 mL for LCF-TV, which uses a lower correction factor. Conversely, the deviation from overestimation was observed to be 7.18 ± 11.39 mL for CU-TV.

Patients who underwent surgery for benign indications (*p* < 0.001) or compressive symptoms (*p* = 0.003) and those with suppressed TSH values (*p* = 0.001) displayed significantly larger discrepancies between US-TV and IO-TV calculations. In contrast, factors like age (*p* = 0.350) and gland nodularity did not significantly influence the discrepancies (*p* = 0.699). Other factors such as sex (*p* = 0.174), BMI (*p* = 0.939), Tg (*p* = 0.128), anti-Tg (*p* = 0.270), anti-TPO (*p* = 0.112), Graves’ disease (*p* = 0.369), Hashimoto’s disease (*p* = 0.432), and the size of the dominant nodule (*p* = 0.285) did not significantly affect the discrepancy between US-TV and IO-TV. Multiple linear regression analysis revealed that only patients who underwent surgery for benign indications (*p* = 0.005) and those with suppressed TSH levels (*p* = 0.013) were found to be significantly related to the difference between US-TV and IO-TV.

An analysis of patient volume distributions according to diagnostic groups and postoperative complications revealed that benign cases exhibited significantly higher IO-TV and US-TV measurements than malignant cases, with a significantly greater amount of underestimation between IO-TV and US-TV measurements. Conversely, it was noted that patients with or without postoperative hypoparathyroidism exhibited comparable underestimation amounts and volume distributions (Table 4).

### 3.4. Thyroid Specimen Weight to Volume Ratio Analysis

When comparing the weight and volume of the specimens, 1 mL of tissue equaled 1.17 g in four patients with Graves’ disease without nodules, 1.07 g in twelve patients with Graves’ disease, and 1.13 g in the whole patient group.

## 4. Discussion

To the best of our knowledge, our study is the only one in the current literature from the last decade that takes into account recent technological advancements in ultrasonography and compares ultrasonographic thyroid volume measurement with specimen volume. As the largest cohort in the literature to date, our findings suggest that ultrasonographic measurements of thyroid volume underestimate the actual volume. However, the deviation in measurements was not as large as the current literature might suggest. Even though the ellipsoid formula produces more accurate results than other proposed formulas, the deviation in measurements increases with higher thyroid volumes. Contrary to the current literature, our data do not show gland nodularity affecting measurement deviations. However, factors such as suppressed TSH levels, benign diagnoses, and the presence of compressive symptoms—which could indicate a larger thyroid volume—significantly influenced the measurement deviations.

Calculating thyroid volume is crucial in various medical disciplines like selecting patients for minimally invasive or endoscopic thyroid surgeries, adjusting doses for thyroid suppression therapy, and radioactive iodine therapy. Few studies in the literature discuss the reliability of this calculation using the ellipsoid formula, and most compare ultrasonographic volumes with other imaging methods like 2D–3D CT or MRI [3,9,15,16]. However, most of these studies do not compare calculated volumes with the actual thyroid volume, which can be determined intraoperatively, leaving the true accuracy of calculated sonographic volumes uncertain.

Our study found significant differences between ultrasonography-based calculated thyroid volume calculations and intraoperatively measured thyroid volumes (IO-TV). Whereas the US-TV and LCF-TV calculations showed underestimation, the formula proposed by Trimboli et al. (CU-TV) showed overestimation [10]. The studies in the literature that compare sonographic calculations with actual thyroid volumes mainly report underestimations of the calculated sonographic volumes. For example, Trimboli et al. found underestimation in 77% of the 92 patients included in their study, which had the second largest thyroid volume range reported after our study. The mean volume distribution of patients in this study was 23.9 ± 14.8 (5.8–85) mL for US-TV and 33.4 ± 20.1 (9.7–105) mL for specimen volume and a limited number of patients with a thyroid volume >60 mL were included [10]. Another study by Miccoli et al., with a sample size of 101 patients, the second largest sample size in the literature after our study, found that ultrasonographic volume measurement resulted in underestimation for 89 (88.1%) patients and overestimation for 7 (6.9%) patients. Only patients with ultrasonographic volumes less than 50 mL were included and the average ultrasonographic volume was 28.3 (7–50) mL, with a mean specimen volume of 36.2 (7–76) mL in this study. When patients were divided into two groups based on the 25 mL threshold in Miccoli’s study, the underestimation rate was higher in the low-volume group and the authors suggested that the consistency of ultrasonographic volume measurement improves with higher volumes [12]. Ruggieri et al. showed that ultrasonographic volume measurement led to 77% underestimation in a group of 53 patients with a mean ultrasonographic volume of 14.4 ± 5.9 mL and a mean specimen volume of 21.7 ± 10.3 mL [11]. Lee et al. compared volume measurements using 2D ultrasound with those from 2D and 3D computed tomography against specimen volume measurements. The mean specimen volume of the 47 patients in Lee et al.’s study was 18.6 mL (min-max range = 8–33 mL), and 2D ultrasound volumes were reported as 19.3 mL (min-max range = 9.2–34.9 mL). They found that all calculated measurements exceeded the specimen volume [15]. All the aforementioned studies primarily included patients with small thyroid volumes, largely aimed at patient selection for minimally invasive thyroid surgery. This makes it challenging to draw conclusions about large thyroid volumes. In this regard, our study, with a volume range of 4–130 mL and a larger sample size of 120 patients, is thought to represent a more realistic clinical population than other studies in the literature. Additionally, we believe that the smaller volume discrepancy between ultrasonographic and specimen measurements in our study compared to other studies in the literature can be ascribed to our higher average thyroid volume and broader volume range, as well as advancements in ultrasonography technology, as our study is the only one conducted in this domain over the past decade.

In our study, the US-TV calculations made by using the ellipsoid formula exhibited a strong correlation with IO-TV. The mathematical formula based on the regression analysis of our results was determined to be IO-TV = 1.522 + 1.047× (US-TV). A formula with a larger correction coefficient was suggested by Trimboli et al. [10]. When using the formula recommended by Trimboli et al. in our study group, we observed an average overestimation of 7.18 ± 11.39 mL (compared to −2.75 ± 10.65 mL in US-TV samples from our study). Furthermore, we found that when the correction factor 0.524, employed in the ellipsoid formula, is adjusted to 0.479, as proposed by Brunn et al., the calculated thyroid volume LCF-TV values intensified the underestimation amounts, which were previously −2.75 ± 10.65, to −4.98 ± 10.98 [8]. A similar observation was made in a study by Shabana et al., where thyroid volumes measured using multidetector computed tomography (MDCT) and ultrasonography were compared with varying correction factors in patients without thyroid pathology [17]. In that study, where the actual thyroid volume was taken as the measurement made using MDCT, the most suitable correction factor was determined to be 0.529, a value very close to the ellipsoid formula correction factor. It can be postulated that all those formulae produce consistent results within the sample group for which they were designed, but this consistency might not hold outside the original sample group. The ellipsoid formula remains the most consistent and reliable method for ultrasonographic volume measurement. The margin of error associated with the ellipsoid formula can be minimized by refining the measurements to better approximate the actual thyroid volume.

In our study, we found that the deviation in ultrasonographic measurements increased with larger thyroid volumes. We also observed that patients with benign diagnoses, compression symptoms, and suppressed TSH values had significantly higher deviation amounts. It appears that a larger thyroid volume might be the common denominator among those factors. An increase in thyroid volume has been noted in cases of benign pathology and compression symptoms in our study. In a study by Guo et al., low TSH levels were associated with higher thyroid volumes, which is consistent with the results of our study [18]. Obtaining precise measurements with ultrasound could be a challenge, particularly with high thyroid volumes. Ultrasonographic volume measurements are employed in longitudinal, transverse, and anteroposterior diameters using two-dimensional images. The linearity of these axes changes when measured as planes, especially as the volume of the thyroid gland increases and the plane begins to angle. Supporting this notion, 3D CT offers volume measurements closer to specimen volumes than 2D ultrasound and 2D CT [15]. The ellipsoid formula calculation is expected to yield more accurate results if the probe is directed parallel to this angulation during measurement. Consequently, the underestimation rate might decrease further due to the more consistent volume calculations made in this way. 

Gland nodularity, in contrast to other studies, was found to have no significant effect on the deviation rate of US-TV measurements in our study. Ruggieri et al. reported that gland multinodularity and nodular involvement of the isthmus significantly affected the underestimation of ultrasonographic thyroid volume measurements [11]. Twenty-one of the fifty-three patients included in this study had isthmus nodularity, and the mean volumes of patients with both multinodularity and isthmus nodularity were greater than the mean volumes of all study participants. As the patients in this study underwent minimally invasive thyroidectomy, their thyroid volumes were within the low-normal range. Patients with isthmus nodules or features like retrosternal extension, cyst aspiration, or damage to the integrity of the specimen that might cause errors in volume measurements were excluded from our study and the isthmus volume was not included in the calculation of the ellipsoid formula. Therefore, we believe the observed underestimation rate in our study, which differs from the results reported by Ruggieri et al., primarily stems from isthmus nodularity and the presence of patients with higher volumes in our study [11].

The precise assessment of thyroid volume is considered crucial not only for selecting optimal surgical treatment strategies, but also for guiding nonsurgical therapies. In the treatment of Graves’ disease, measuring thyroid volume is crucial for determining the dose of radioactive iodine therapy [4]. For this purpose, 1 g of thyroid tissue is deemed equivalent to 1 cm^3^ of thyroid volume [9,15]. In our study, we observed that in patients with Graves’ disease, 1 cm^3^ of tissue corresponds to 1.17 g in four patients without nodularity and 1.07 g in twelve patients with nodularity. Though this discrepancy appears minimal, a variation of roughly 17% per mL could be significant in terms of treatment complications. However, validation with studies using a larger patient series is necessary for this recommendation to attain clinical significance.

Accurate thyroid volume determination aids in understanding patients’ susceptibility to postoperative complications. In a study by Dong et al., preoperative ultrasonographic thyroid volume was higher in patients undergoing difficult thyroidectomies, which was associated with higher complication rates [19]. In another study that evaluated the relationship between the external branch of the superior laryngeal nerve and thyroid volume according to the Cernea classification, it was reported that the type-2 anatomy with the highest risk of nerve damage was more frequently observed in thyroid glands larger than 50 mL [20]. In a study investigating the relationship between retrosternal extension of the thyroid gland and the need for an extracervical approach, thyroid volume showing mediastinal extension exceeding 162 cm^3^ from the thoracic inlet was a determinant factor for an extracervical approach for thyroidectomy [21]. In a retrospective study of 392 Graves’ disease patients, it was reported that elevated T3 levels were effective in both larger thyroid sizes and the difference between preoperative ultrasonographic volume and pathological sizes. Additionally, suppressed TSH levels were found to play a role in the development of postoperative hypocalcemia [22]. The low complication rates observed in our study patients and the differences in study design preclude a direct comparison of these factors. 

The strengths of this study, compared to others that used specimen volume, include it being the largest in number and having the widest volume range cohort in the literature. It demonstrates data reflecting recent technological developments in ultrasound and includes patients who underwent a complete gland dissection by a single surgeon. Among the study’s limitations is the fact that ultrasonographic volume measurement, being a 2D method, might be insufficient to capture the full benefits that 3D approaches can offer in clinical practice. Secondly, due to the retrospective design of the study, we could not evaluate the effects of factors such as neck thickness and length on the margin of error in patients. These factors have the potential to influence ultrasonographic measurements.

## 5. Conclusions

Our study findings indicate that ultrasonographic measurements of thyroid volume tend to underestimate the actual thyroid volume. This deviation, which is less pronounced than what the current literature suggests, seems to be caused by factors other than nodularity, which increase the volume of the thyroid gland. We recommend the continued use of the ellipsoid formula as the most valid and accurate method for thyroid volume measurement. However, incorporating 3D imaging techniques, such as CT or MRI, may minimize the margin of error, especially in situations where precise measurements are crucial for large-volume thyroid glands. To enhance the accuracy of ultrasonographic volume measurements further, we believe that three-dimensional analysis and adjustments made during measurements can reduce the elevated margin of error, especially in the context of large-volume thyroid glands.

## Figures and Tables

**Figure 1 jcm-12-06619-f001:**
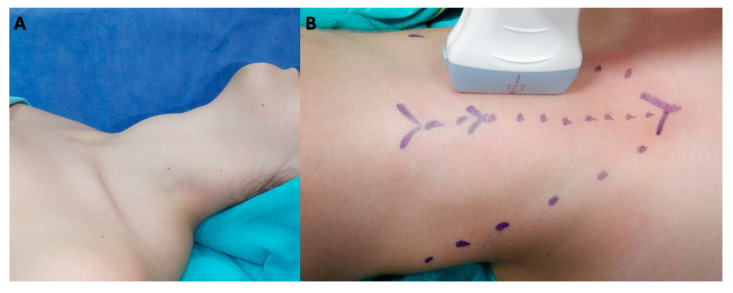
Pre-incision Positioning of Patients under Anesthesia on the Operating Table (**A**) and Ultrasonographic Evaluation of the Thyroid Gland (**B**).

**Figure 2 jcm-12-06619-f002:**
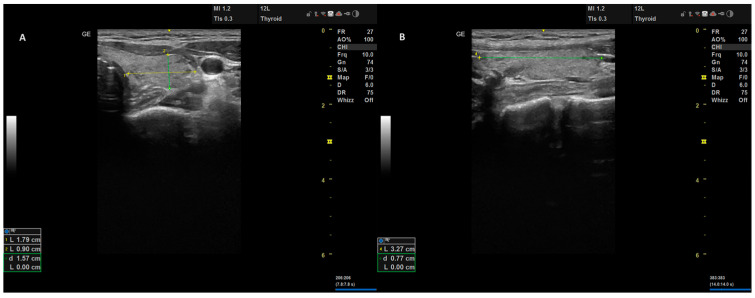
Measurement of Mediolateral and Anteroposterior Dimensions (**A**) and Craniocaudal Dimension (**B**) During Ultrasonographic Assessment.

**Figure 3 jcm-12-06619-f003:**
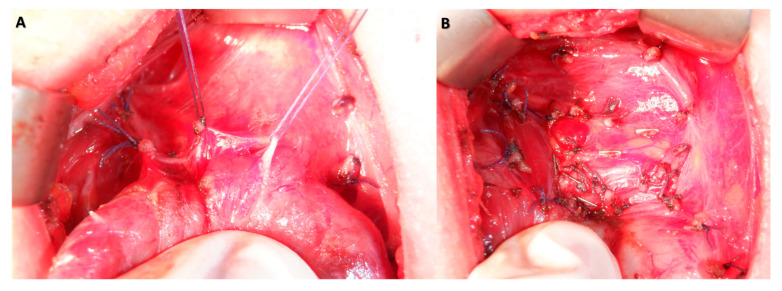
Application of Capsular Dissection Technique during Surgical Procedure (**A**) and Surgical Field View Demonstrating the Absence of Macroscopic Residual Thyroid Tissue following Total Thyroidectomy (**B**).

**Figure 4 jcm-12-06619-f004:**
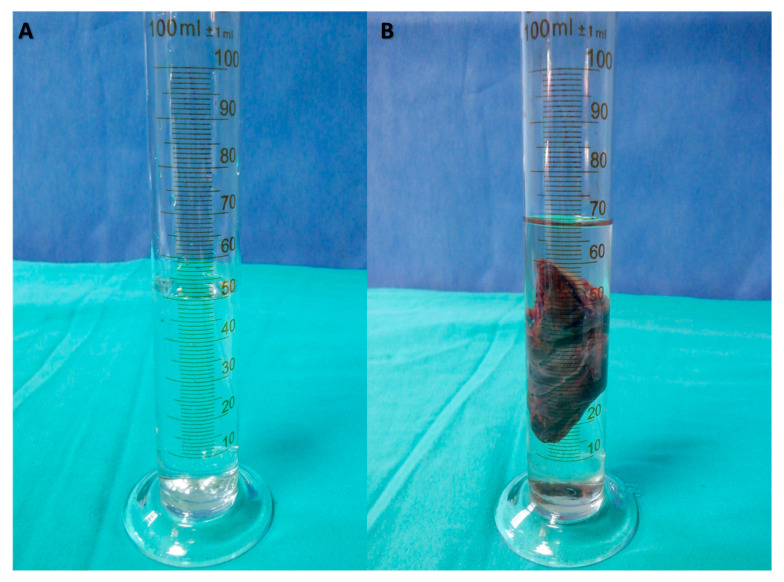
Placement of an Adequate Amount of Saline in a Graduated Cylinder Appropriate to Submerge the Specimen (**A**) and Measurement of Specimen Volume in Milliliters according to Archimedes’ Principle after Submerging the Specimen in Saline (**B**).

**Figure 5 jcm-12-06619-f005:**
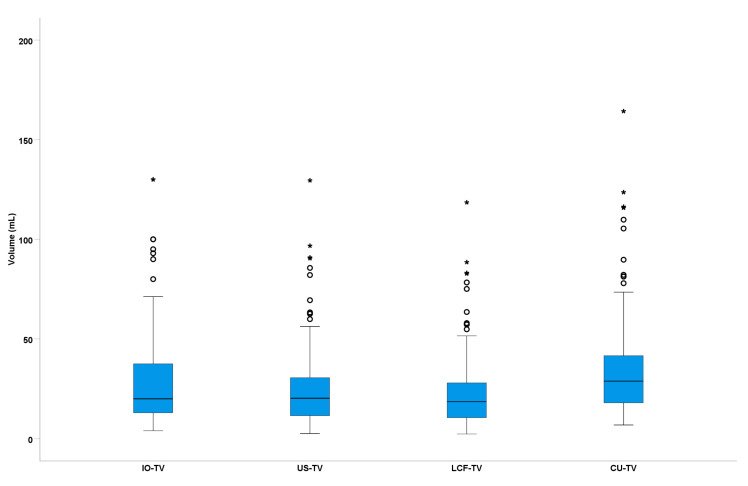
Box-and-whisker plot comparing IO-TV and calculated thyroid volumes. O denotes outliers and * indicates extreme values.

**Figure 6 jcm-12-06619-f006:**
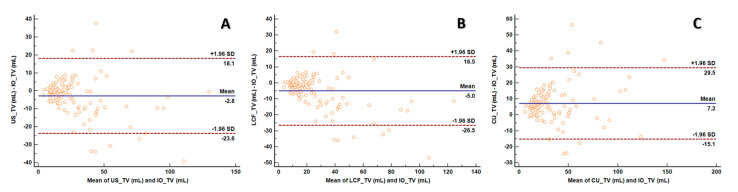
Bland–Altman Plots comparing IO–TV and various calculated ultrasonographic thyroid volumes. A–C, the plots show volume measurements of US–TV (**A**), LCF–TV (**B**), and CU–TV (**C**) against IO–TV. IO–TV: intraoperative (specimen) thyroid volume, US–TV: ultrasonographic thyroid volume, LCF–TV: lower correction factor thyroid volume, CU–TV: calculated ultrasound thyroid volume.

**Figure 7 jcm-12-06619-f007:**
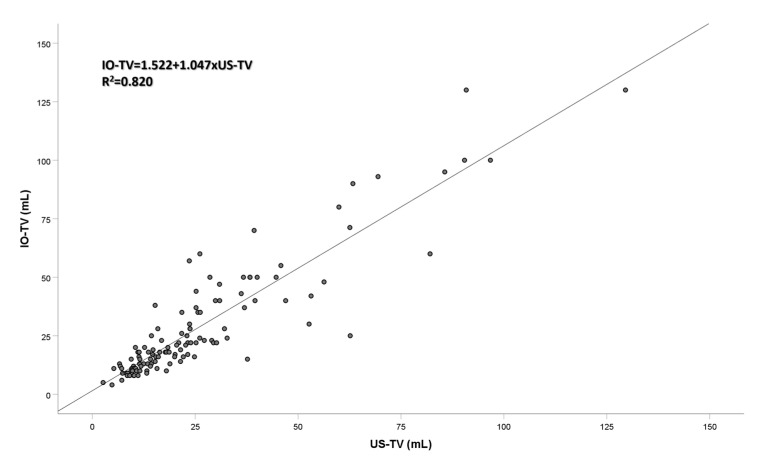
Linear Regression Applied Between IO-TV and US-TV. IO-TV: Intraoperative (Specimen) Thyroid Volume, US-TV: Ultrasonographic Thyroid Volume.

**Table 1 jcm-12-06619-t001:** Descriptive Statistics and Volume Measurements of the Patients.

Demographic Data (*n* = 120)	
Age, years (*mean ± SD*)	46.02 ± 13.20
Sex, *F*/*M*, *n* (%)	99/21 (82.5/17.5)
BMI (*mean ± SD*)	27.55 ± 5.31
**Preoperative Diagnostic Evaluation**	
Graves’ Disease, (*Yes*/*No*), *n* (%)	12/108 (10.0/90.0)
Multinodular Goiter, (*Yes*/*No*), *n* (%)	92/28 (76.7/23.3)
Unilateral, *n* (%)	16 (13.3)
Bilateral, *n* (%)	76 (63.3)
Suspicion for Malignancy, (*Yes*/*No*), *n* (%)	100/20 (83.3/16.7)
Size of Dominant Nodule, mm (*mean ± SD*)	18.72 ± 12.86
Presence of Compressive Symptoms, (*Yes*/*No*), *n* (%)	18/102 (15.0/85.0)
**Preoperative Laboratory Parameters**, (*mean ± SD*)	
TSH, µIU/mL (0.38–5.33)	1.75 ± 1.60
Thyroglobulin, ng/mL (1.15–50.03)	95.31 ± 225.82
Anti-Tg Ab, IU/mL (0–4)	117.17 ± 384.22
Anti-TPO Ab, IU/mL (0–9)	64.44 ± 115.88
**Pathology**, *n* (%)	
Benign	37 (30.8)
Malignant	83 (69.2)
Presence of Autoimmune Thyroiditis, (*Yes*/*No*) *n* (%)	50/70 (41.7/58.3)
**Volume Measurements**, mL (*mean ± SD*)	
IO-TV	28.90 ± 24.96
US-TV	26.15 ± 21.58
LCF-TV	23.92 ± 19.74
CU-TV	36.09 ± 26.76

BMI: Body Mass Index, TSH: Thyroid-Stimulating Hormone, anti-Tg Ab: Anti-Thyroglobulin Antibody, anti-TPO Ab: Anti-Thyroid Peroxidase Antibody, IO-TV: Intraoperative (Specimen) Thyroid Volume, US-TV: Ultrasonographic Thyroid Volume, LCF-TV: Lower Correction Factor Thyroid Volume, CU-TV: Calculated Ultrasound Thyroid Volume.

**Table 2 jcm-12-06619-t002:** Intraoperative Volume vs. Calculated Ultrasonographic Volume Measurements for Patients, Along with Estimation Errors.

Methods Being Compared	Mean ± SD *	95% CI for Mean	Median (25th–75th Percentile)	*p*-Values
US-TV–IO-TV (mL)	−2.75 ± 10.65	(−4.68–−0.83)	−0.79 (−6.87–2.09)	**0.005**
LCF-TV–IO-TV (mL)	−4.98 ± 10.98	(−6.96–−3.00)	−2.28 (−10.16–0.60)	**<0.001**
CU-TV–IO-TV (mL)	7.18 ± 11.39	(5.12–9.24)	6.25 (1.52–11.46)	**<0.001**

* Estimation error was calculated by subtracting the value of the reference method (IO-TV) from the compared methods (US-TV, LCF-TV, and CU-TV). IO-TV: Intraoperative (Specimen) Thyroid Volume, US-TV: Ultrasonographic Thyroid Volume, LCF-TV: Lower Correction Factor Thyroid Volume, CU-TV: Calculated Ultrasound Thyroid Volume. Statistically significant comparisons presented as bold.

**Table 3 jcm-12-06619-t003:** Correlation Matrix for Thyroid Volume Calculation Methods with Specimen Volumes.

	IO-TV vs. US-TV	IO-TV vs. LCF-TV	IO-TV vs. CU-TV
**CCC (ρ_c_) ^a^**	0.890	0.860	0.869
**95% CI for CCC**	0.848–0.920	0.813–0.896	0.821–0.905
**Pearson Correlation Coefficient (r)**	0.905	0.905	0.905
**Bias Correction Factor (C_b_)**	0.983	0.950	0.960
**McBride’s Strength of Agreement Criteria ^b^**	Poor	Poor	Poor
** *p* **	**<0.001**	**<0.001**	**<0.001**

^a^ Calculated using the following formula: ρc = ρ Cb; ^b^ McBride’s strength of agreement criteria for CCC (ρc) are as follows: <0.90: poor; 0.90–0.95: moderate; 0.95–0.99: substantial; >0.99: almost perfect. CCC: concordance correlation coefficient. Statistically significant comparisons presented as bold.

**Table 4 jcm-12-06619-t004:** Analysis of patient volume distributions and estimation errors according to diagnostic groups and postoperative complications.

	*n*	IO-TV (mL)	*p*1 *	US-TV (mL)	*p*2 **	(US-TV)–(IO-TV) (mL) †	*p*3 ***
**Pathology**							
Benign	37	42.00 (22.00–60.00)	**<0.001**	26.13 (19.99–53.15)	**<0.001**	−5.34 (−13.28–1.09)	**0.006**
Malignant	83	17.00 (11.00–25.00)		15.72 (10.76–25.20)		−0.46 (−4.66–2.15)	
**Postoperative Hypoparathyroidism**							
None	93	20.00 (13.00–40.00)	0.956	20.48 (11.81–32.11)	0.545	−0.52 (−6.92–2.44)	0.261
Protracted	21	19.00 (16.00–30.00)		20.99 (11.56–23.56)		−1.02 (−5.34–1.19)	
Transient	6	21.00 (11.00–30.00)		14.51 (10.54–23.63)		−4.38 (−10.64–−2.35)	
Permanent	0	-		-		-	

Data presented as Median (percentile 25th–percentile 75th). * *p*1: Comparison of IO-TV measurements between groups, ** *p*2: comparison of US-TV measurements between groups, *** *p*3: comparison of (US-TV)–(IO-TV) amounts between groups, †: estimation error was calculated by subtracting the value of the reference method (IO-TV) from the compared methods (US-TV). IO-TV: intraoperative (specimen) thyroid volume, US-TV: ultrasonographic thyroid volume. Statistically significant comparisons presented as bold.

## Data Availability

The data presented in this study are available on request from the corresponding author. The data are not publicly available due to the need to protect patients’ confidential information.

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
