# Peer review of "Unveiling the Accuracy of Ultrasonographic Assessment of Thyroid Volume: A Comparative Analysis of Ultrasonographic Measurements and Specimen Volumes"

_jcm, 2023, doi:10.3390/jcm12206619_

Round 1

Reviewer 1 Report

The manuscript is great, with minor revision necessary, as yellow highlighted text and comments on the attached PDF text.

The addition of photos of ultrasound procedures , intraoperative surgery, and technical measurement of specimen volume will enhance the manuscript.

References list should be optimized with more recent literatures added

Thank you

The language is great.

Reviewer 2 Report

Interesting study providing new data regarding the evaluation of thyroid nodules. Consider editing the discussion to be more concise, would help with readability. Unclear the additional benefit of comparing the CP-TV to IO-TV, as this is not used to calculate TV pre-operatively. Elminating this could help with clarity of results and discussion.

Reviewer 3 Report

The authors reported a study comparing thyroid ultrasound size versus histological specimen.
The study remains interesting as it is pilot with recent imaging techniques.
Some remarks:
- The authors have placed the 15 references of their article in the introduction: the discussion merely repeats the same references. The authors should spare the intro a little more, and improve the references in the discussion.

- How was the ultrasound diagnosis of malignancy made? when it was a nodule, the recommendations suggest a cytopunction: how was the selection of patients going directly to surgery made?

The authors could place EuTirads score in there to argument their indications ? ( EuT V < 10 mm ? )

- Were there any diagnoses of thyroid infections: abscesses, tuberculosis?

- Were calcitonin levels measured?

- Could you give us more information on post-operative complications, ultrasound thyroid volume and histology?

- And what are the practical consequences of this assessment? Are there any measures to be taken when assessing a cut-off in ultrasound thyroid size that increases the risk of malignancy.
